

# A comparative study on leaf anatomy and photosynthetic characteristics of different growth stages of *Horsfieldia hainanensis*

Jianwang Xu[1,2], Jianmin Tang[2], Haolong Jiang[2], Rong Zou[2] and Xiao Wei[1,2]

[1] College of Pharmacy, Guilin Medical University, Guilin, Guangxi, China
[2] Guangxi Institute of Botany, Chinese Academy of Sciences, Guilin, Guangxi, China

Corresponding authors
Rong Zou, zr@gxib.cn
Xiao Wei, wx@gxib.cn

## ABSTRACT

The rare and endangered wild plant, *Horsfieldia hainanensis*, has been listed as a second-level key protected plant in China. Currently, its habitat is severely damaged, and the population has dramatically declined, necessitating urgent intervention for protection. In this study, the aim was to explore the correlations and differences from the perspectives of photosynthetic characteristics and leaf structure, providing scientific references for *in-situ* conservation and *ex-situ* cultivation. The results revealed the following: (1) The maximum net photosynthetic rate ($P_{max}$) and light saturation point (LSP) of mature trees were significantly higher than those of seedlings, while the light compensation point (LCP), $CO_2$ compensation point (CCP), and $CO_2$ saturation point (CSP) were significantly lower in seedlings. (2) The average daily net photosynthetic rate of mature trees was significantly higher than that of seedlings. When both mature trees and seedlings exhibited a "midday depression" phenomenon, accompanied by an increase in intercellular $CO_2$ concentration ($C_i$), it indicated that the "midday depression" was caused by non-stomatal limiting factors. (4) Both mature trees and seedlings showed peak values of water use efficiency (*WUE*) under low light conditions. Mature trees had smaller upper and lower epidermis thickness but larger leaf thickness, and their leaf structure, characterized by well-developed palisade and spongy tissues, conformed to the cellular structure adaptations for low light. Therefore, both were more adapted to low light conditions. (5) The stomatal density (SD) and individual stomatal area (SA) of seedlings were significantly higher than those of mature trees. (6) The total chlorophyll content of mature trees was significantly higher than that of seedlings, while the chlorophyll a/chlorophyll b ratio was significantly lower in mature trees and remained below three in both cases. In summary, the photosynthetic capacity and light adaptability of mature trees are stronger than those of seedlings, but both mature trees and seedlings exhibit shade-tolerant characteristics. For *in-situ* conservation, it is possible to promote the growth and development of seedlings by appropriately employing artificial "windowing" or shading methods based on the actual growth environment of the seedlings. In the case of *ex-situ* cultivation, seedlings should be provided with appropriate shading initially, while ensuring sufficient moisture and $CO_2$ concentration. As the plants grow, the shading intensity can be gradually reduced. Once the plants reach maturity, they have a broader range of light adaptability and can be transplanted to environments with less shading.

## INTRODUCTION

*Horsfieldia hainanensis*, a tall tree of the Myristicaceae family, is primarily found in Hainan and southern Guangxi, China. This species typically inhabits shady, moist forests in valleys and hills at elevations between 400 and 450 m above sea level. It has been listed in both the "List of Extremely Small Population Wild Plants in Guangxi" and the "List of Key Protected Wild Plants in China." Due to its limited geographical distribution, population decline, and rarity, *H. hainanensis* is considered part of an extremely small population. Such populations are critical reservoirs of biological and genetic diversity. Without timely conservation efforts, their unique biological traits and genetic resources may be lost. As a result, protecting plants of extremely small populations is crucial for preserving biodiversity in China.

Photosynthesis is the fundamental process by which green plants utilize light energy to produce and store organic compounds. It is also a vital physiological process closely associated with the growth and development of plants (*Zong et al., 2022*). The competitive advantage of plants in population competition and their distribution range are indirectly influenced by the strength of photosynthetic capacity. By studying plant photosynthesis and its influencing factors, it is possible to optimize environmental conditions during cultivation and breeding processes. This is an important approach in the protection of endangered plants, population restoration, and reintroduction efforts (*John et al., 2007*). For instance, the related research findings indicate significant differences in photosynthetic and photoprotective characteristics between *Nerium oleander* seedlings and mature trees, leading to distinct physiological traits in response to different seasons (*Chondrogiannis et al., 2023*). A comparative analysis of leaf characteristics of seedlings and mature trees of 17 species in tropical rainforests revealed significant differences in photosynthetic characteristics and leaf structures between seedlings and mature trees of all species, which are correlated with their environmental adaptability and growth status (*Houter & Pons, 2012*). Additionally, leaf microstructure and photosynthetic pigment content are crucial methods for observing plant diversity. *Pan et al. (2024a)* found significant differences in photosynthetic characteristics between seedlings and mature trees of *Manglietia aromatica*, primarily attributed to differences in spongy tissue thickness, leaf thickness, and chlorophyll content. From the aforementioned studies, it is evident that understanding the differences in photosynthetic characteristics and physiological structures of the same plant species at different stages is crucial for elucidating their endangerment mechanisms and implementing appropriate conservation measures.

The natural regeneration of *H. hainanensis* is challenging, with low survival rates of seedlings impeding their growth into mature trees, thereby severely affecting population propagation and expansion. In order to investigate the reasons behind these challenges and

implement rational conservation measures for the *H. hainanensis*, researchers have conducted studies on aspects such as reproductive cultivation, genetic diversity, ecological characteristics, and population structure (*He, 2013*; *Chai et al., 2021*; *Jiang et al., 2016*; *Zhong et al., 2018*). However, there is limited information regarding its photosynthetic characteristics and leaf microstructure. This study aims to address the following questions through the measurement and analysis of photosynthetic characteristics, leaf structure features, and photosynthetic pigment content of mature trees and seedlings of the *H. hainanensis*: (1) Are there differences in photosynthetic characteristics between mature trees and seedlings of the *H. hainanensis*? (2) If differences in photosynthetic capacity exist, are these differences related to leaf structure and photosynthetic pigment content? The research results aim to explore the photosynthetic characteristics and differences at different growth stages of the *H. hainanensis*, with the intention of providing a theoretical basis for translocation conservation and population restoration of the species and investigating protective planting methods for the seedlings of the *H. hainanensis*.

## MATERIALS AND METHODS

### Overview of the experimental site

The experimental site is located within the Guangxi Institute of Botany in Guilin, Guangxi. It is situated at approximately 25°01′N latitude and 110°17′E longitude, with an elevation of 180 m. The site falls within the subtropical monsoon climate zone of Central Asia. The region benefit from favorable climatic conditions with abundant sunlight and ample rainfall, it is an ecological habitat suitable for the growth of *H. hainanensis*. The average annual temperature is 19.4 °C, with the hottest month averaging around 28.5 °C and the coldest month averaging only about 8.3 °C. The average annual accumulative precipitation is approximately 1,974 mm, and the average annual relative humidity ranges from 73% to 79%. The average annual sunshine duration is around 1,670 h (*Wu et al., 2000*).

### Materials

The test materials consisted of mature trees and seedlings of *H. hainanensis*. The 12-year-old mature trees were planted in the endangered plant germplasm nursery of the Guangxi Institute of Botany. The experimental seedlings are 1-year-old seedlings cultivated from seeds collected from mature trees in the study. They were planted in plastic pots with an inner diameter of 21 cm and depth of 18 cm, with one seedling per pot. Both mature trees and seedlings are grown in red soil as the soil substrate. Three healthy and vigorous mature trees and seedlings without diseases or pests were selected for the experiments.

### Methods

#### Measurement of photosynthesis-light response curves

Under clear weather conditions, from 8:00 to 13:00 in the morning, the photosynthetic parameters of *H. hainanensis* were measured using a portable Li-6400 photosynthesis system (Li-6400; Li-Cor, Lincoln, NE, USA). Prior to measurement, the leaves were pre-conditioned for 30 min under a light intensity of 600 $\mu mol \cdot m^{-2} \cdot s^{-1}$ (using the built-in
red-blue light source of the instrument) to fully activate the photosynthetic system. The measurement was conducted with an open gas exchange system, with an airflow rate of 0.5 L·min$^{-1}$, leaf temperature set at 28 °C, and $CO_2$ concentration set at 400 μmol·mol$^{-1}$. The light intensity gradient was set at 1,500, 1,200, 1,000, 800, 600, 400, 200, 150, 100, 50, 20, and 0 μmol·m$^{-2}$·s$^{-1}$. Three individuals each of mature trees and seedlings were measured. The photosynthetic-light response curve was plotted with photosynthetic photon flux density (PPFD) as the x-axis and net photosynthesis rate (Pn) as the y-axis. The photosynthetic parameters were fitted to the Pn-PPFD curve using the following equation (*Ye, Yu & Kang, 2012*):

$$Pn = \mathrm{AQY}\frac{1 - \beta \mathrm{PPFD}}{1 + \gamma \mathrm{PPFD}}\mathrm{PPFD} - R_{\mathrm{d}}. \tag{1}$$

In the equation, Pn represents the net photosynthesis rate, AQY is the apparent quantum yield, β and γ are coefficients, PPFD denotes the photosynthetic photon flux density, and $R_{\mathrm{d}}$ stands for dark respiration rate. After conducting a goodness-of-fit test and obtaining satisfactory results, the following formulas were used to calculate the light saturation point (LSP), maximum net photosynthesis rate ($P_{\mathrm{max}}$), and light compensation point (LCP):

$$\mathrm{LSP} = \frac{\sqrt{(\beta + \gamma)/\beta} - 1}{\gamma} \tag{2}$$

$$P_{\mathrm{max}} = \mathrm{AQY}\left(\frac{\sqrt{\beta + \gamma} - \sqrt{\beta}}{\gamma}\right)^2 - R_{\mathrm{d}} \tag{3}$$

$$\mathrm{LCP} = \frac{\mathrm{AQY} - \gamma R_{\mathrm{d}} - \sqrt{(\gamma R_{\mathrm{d}} - AQY)^2 - 4\beta \times \mathrm{AQY} \times R_{\mathrm{d}}}}{2\alpha\beta}. \tag{4}$$

During the extraction of the light response curves, stomatal conductance (Gs) and transpiration rate (Tr) were analyzed at different light intensities. Water use efficiency (WUE = Pn/Tr) was calculated (*Wen, 1997*).

### Measurement of photosynthesis-CO$_2$ response curves

Three healthy plants at each growth stage, including both mature trees and seedlings, previously used for photosynthesis-light response curve measurements were selected for analysis, with each plant replicated three times. Measurements were conducted on the test leaves in the morning at 8:00 after induction. The measurement of photosynthesis-$CO_2$ response curves was conducted using the Li-6400 portable photosynthesis system. The airflow rate was set at 0.5 L·min$^{-1}$, and the leaf temperature was set at 28 °C. Based on the results of the light response curve measurement, the fixed light intensity for mature trees was set at 1,000 μmol·m$^{-2}$·s$^{-1}$, and for seedlings, it was set at 500 μmol·m$^{-2}$·s$^{-1}$. The $CO_2$ concentration gradient was set at 400, 300, 200, 150, 100, 50, 400, 400, 600, 800, 1,000, 1,200, 1,500, and 2,000 μmol·mol$^{-1}$ (controlled using $CO_2$ cylinders). During the measurement, a balance of 150–180 s was maintained at each $CO_2$ concentration, and the

net photosynthesis rate ($Pn$) at different $CO_2$ concentrations was automatically recorded by the system. The $Pn$-$C_i$ curve was fitted and plotted using a rectangular hyperbolic correction model. Net photosynthesis rate ($Pn$), $CO_2$ compensation point (CCP), $CO_2$ saturation point (CSP), and potential maximum net photosynthesis rate ($A_{max}$) were calculated using the following formulas.

$$Pn = \alpha \frac{1 - \beta C_i}{1 + \gamma C_i} C_i - R_p \tag{5}$$

$$CCP = \frac{\alpha - \gamma R_p - \sqrt{(\alpha - \gamma R_p)^2 - 4\alpha\beta R_p}}{2\alpha\beta} C_i - R_p \tag{6}$$

$$CSP = \frac{\sqrt{(\beta + \gamma)/\beta} - 1}{\gamma} \tag{7}$$

$$A_{max} = \alpha \left( \frac{\sqrt{\beta + \gamma} - \sqrt{\beta}}{\gamma} \right)^2 - R_p. \tag{8}$$

In the equation, $C_i$ represents the intercellular $CO_2$ concentration, $\alpha$ is the initial carboxylation efficiency of the $CO_2$ response curve, $\beta$ and $\gamma$ are coefficients, and $R_p$ is the rate of photorespiration.

### Measurement of diurnal variation in photosynthesis

Three healthy plants at each growth stage, including both mature trees and seedlings, previously used for photosynthesis-light response curve measurements were selected for analysis, with each plant replicated three times. The parameters of photosynthetic diurnal changes were measured using the Li-6400 portable photosynthesis system. Measurements were taken at intervals of 1.5 h between 8:30 and 17:30 Beijing time. The following photosynthetic parameters and environmental factors were measured: net photosynthesis rate ($Pn$, µmol·m$^{-2}$·s$^{-1}$), transpiration rate ($Tr$, mmol·m$^{-2}$·s$^{-1}$), stomatal conductance ($Gs$, mol·m$^{-2}$·s$^{-1}$), intercellular $CO_2$ concentration ($C_i$, µmol·mol$^{-1}$), stomatal limitation ($L_s = 1 - C_i/C_a$), water use efficiency ($WUE = Pn/Tr$, µmol·mmol$^{-1}$). Additionally, environmental factors such as photosynthetically active radiation (PAR, µmol·m$^{-2}$·s$^{-1}$), air temperature ($T_a$, °C), and relative humidity ($RH$, %) were recorded.

### Measurement of leaf anatomy parameters

Three healthy and disease-free leaves, located 2–3 pairs down from the top, were separately collected from three mature trees and three seedlings, resulting in a total of three samples per treatment. Small sections measuring 1 cm × 1 cm were excised and fixed in 70% Formalin-Acetic Acid-Alcohol (FAA) fixative solution for 24 h. Subsequently, conventional paraffin sectioning technique (*Li et al., 2019*) was employed to prepare slides, which were then observed and photographed using a Nikon Eclipse E100 optical microscope. Six slides were obtained for each treatment, with three fields of view captured per slide. Additionally, the leaf anatomy parameters were measured using caseviewer

software, including leaf thickness, upper epidermis thickness, lower epidermis thickness, palisade parenchyma thickness, and spongy parenchyma thickness. Furthermore, the ratios of palisade to spongy tissue, relative palisade parenchyma thickness, and relative spongy parenchyma thickness were calculated as follows: (1) Palisade-to-spongy tissue ratio = palisade parenchyma thickness/spongy parenchyma thickness; (2) Relative palisade parenchyma thickness = palisade parenchyma thickness/leaf thickness; (3) Relative spongy parenchyma thickness = spongy parenchyma thickness/leaf thickness.

### Measurement of leaf stomatal structure

By comparing the leaf stomatal structure parameters of mature trees and seedlings, we further analyzed the differences in stomatal structure between the upper and lower epidermises of leaves of mature trees and seedlings. We adopted the same sampling method as that for determining leaf anatomical structure. A total of 18 leaves from mature trees and seedlings were picked. For each leaf, three 1 cm × 1 cm samples were cut from both the upper and lower epidermises. For each sample, 2.5% glutaraldehyde solution was used for fixation, followed by rinsing with phosphate buffer. Dehydration, critical point drying and gold plating were carried out in sequence. The VEGA3 TESCAN vacuum electron scanning electron microscope was used for photographing and observation. Ten fields of view were randomly observed for each sample. Finally, Axiovision was used to measure stomatal length, stomatal width, stomatal density and single stomatal area.

### Measurement of photosynthetic pigment content

The leaf samples used for the photosynthetic measurements were individually weighed to 0.5 g, excluding the veins. They were then cut into small pieces and transferred to 25 mL volumetric flasks. Next, 95% ethanol was added to each flask to reach the volume mark. The flasks were placed in dark conditions for 24 h to allow for pigment extraction. Afterward, the absorbance of the extraction solution was measured at wavelengths of 665 nm, 649 nm, and 470 nm. This process was repeated three times for each sample. Using the following formulas, the content of chlorophyll a (Chl a), chlorophyll b (Chl b), total chlorophyll (Chl a+b), and carotenoids (Car) was calculated, along with the ratio of Chl a to Chl b (Chl a/b) and the ratio of carotenoids to total chlorophyll content (Car/Chl a +b) (*Li et al., 2020b*):

$$\text{Chl a} = 13.95A_{665} - 6.88A_{649} \tag{9}$$

$$\text{Chl b} = 24.96A_{649} - 7.32A_{665} \tag{10}$$

$$\text{Chl a} + \text{b} = 18.08A_{649} + 6.63A_{665} \tag{11}$$

$$\text{Car} = \frac{1000A_{470} - 2.05\text{Chl a} - 114.8\text{Chl b}}{245}. \tag{12}$$

### Data analysis

The mean and standard deviation data of photosynthetic characteristic parameters, leaf structure parameters, and photosynthetic pigment content were calculated using Excel

2016. A t-test was performed using SPSS 26.0 (IBM, Armonk, NY, USA), graphs were plotted using Origin 2022 software, and the photosynthetic parameters of the photosynthetic response curve were fitted and calculated using the rectangular hyperbolic correction model from the Photosynthesis Calculation 4.1.1 software (*Ye, 2010*).

## RESULTS

### Diurnal variation of photosynthetic parameters

From Fig. 1, it can be observed that the *Pn* of both mature trees and seedlings of *H. hainanensis* showed a low value at 13:00 Beijing time, exhibiting a typical "midday depression" phenomenon, with the highest *Pn* value at 10:00. The diurnal variations of $L_s$ and *WUE* were similar to *Pn*, showing low values at 13:00 and relatively high values at 11:30 and 14:30, then dropping to low values at 13:00, with the difference that *WUE* remained at relatively high values before 11:30. The $C_i$ showed high values at 8:30, 13:00, and 17:30, with a low value at 11:30. The diurnal variation trends of $G_s$ and *Tr* were similar. $G_s$ of mature trees showed a peak at 11:30 and a decrease at 14:30, while seedlings exhibited a unimodal trend with the highest value at 10:00 and the lowest value at 17:30. Table 1 indicates that the daily average net photosynthetic rate and daily average stomatal limitation of mature trees were greater than those of seedlings, showing extremely significant differences ($P < 0.01$). On the other hand, the daily average intercellular $CO_2$ concentration and daily average water use efficiency of seedlings were higher than those of mature trees, showing significant differences ($P < 0.05$). There were no significant differences ($P > 0.05$) in the daily average stomatal conductance and daily average transpiration rate between mature trees and seedlings.

### Diurnal variation of environmental factors

The diurnal variation of environmental factors, as shown in Fig. 2, revealed that temperature (*Ta*) and photosynthetically active radiation (PAR) exhibited a unimodal trend with an initial increase followed by a decrease. *Ta* reached its peak at 13:00 (Beijing time), with a value of 24.384 °C. PAR reached its peak at 10:00, with a value of 7.492 $\mu mol \cdot m^{-2} \cdot s^{-1}$. *RH* reached its lowest point at 13:00, with a value of 37.955%, which may be related to the ambient temperature at that time. It then reached its highest value at 17:30, measuring 51.183%.

### Response of photosynthetic physiological indices to light intensity

The determination coefficients ($R^2$) for the fitting of the photosynthesis-light response curves in mature trees and seedlings of *H. hainanensis* were both above 0.95, indicating a good fitting effect (Fig. 3). As the photosynthetic photon flux density (PPFD) increased, the *Pn* of mature trees gradually increased and then stabilized, while the *Pn* of seedlings initially increased and then showed a declining trend. The *Pn* of the two groups started to diverge after a PPFD of 200 $\mu mol \cdot m^{-2} \cdot s^{-1}$. The stomatal conductance ($G_s$) of mature trees increased with increasing light intensity, while the $G_s$ of seedlings showed an initial increase followed by a flattening trend. Differences in $G_s$ between the two groups gradually emerged after a PPFD of 400 $\mu mol \cdot m^{-2} \cdot s^{-1}$, indicating that mature trees possess a stronger

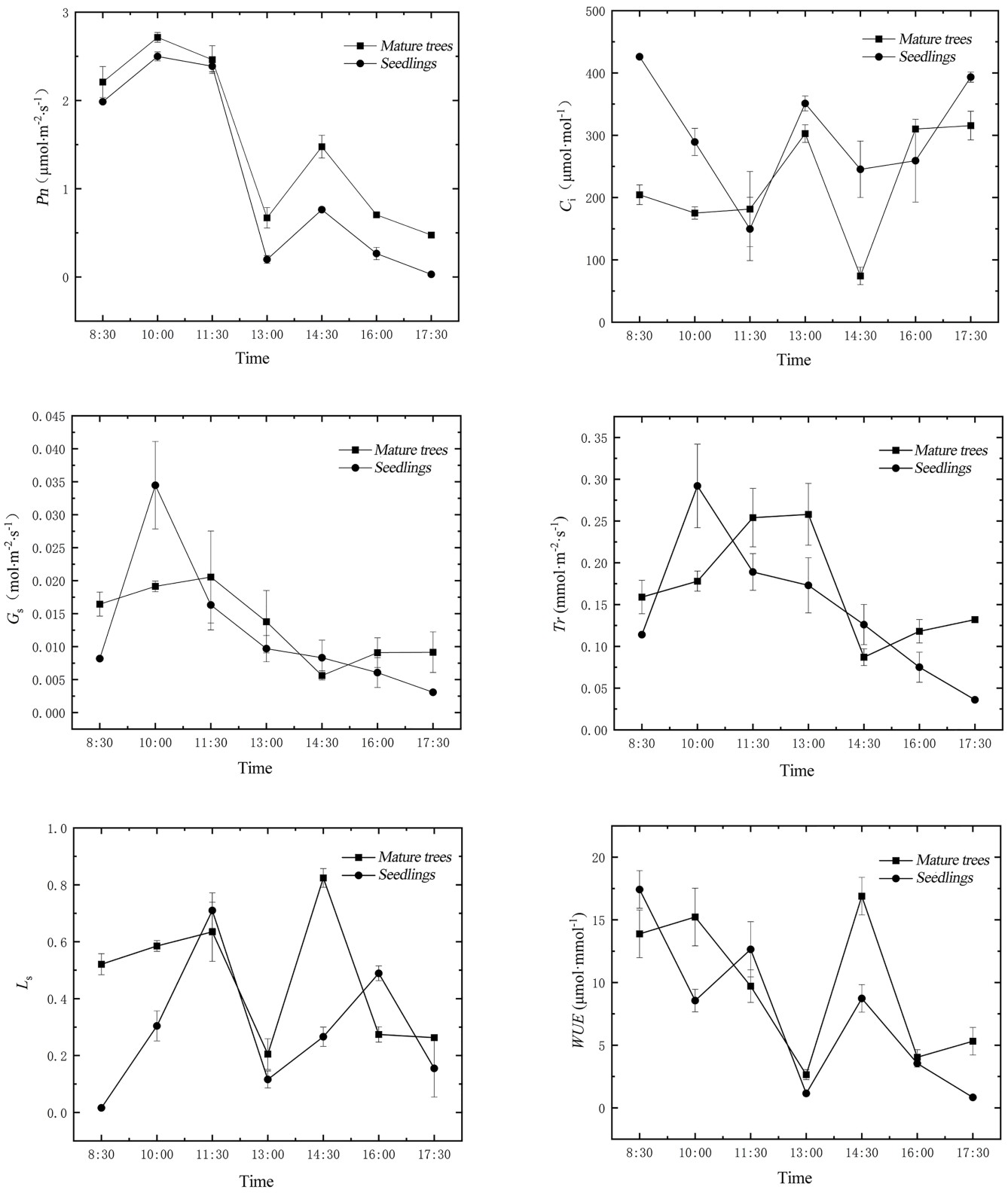

**Figure 1 Diurnal variation of photosynthetic parameters in leaves of mature trees and seedlings of *H. hainanensis*.**

**Table 1 Daily average values of photosynthetic parameters in leaves of mature trees and seedlings of *H. hainanensis*.**

| Type | $P_n$ (μmol·m⁻²·s⁻¹) | $G_s$ (mol·m⁻²·s⁻¹) | $C_i$ (μmol·mol⁻¹) | $Tr$ (mmol·m⁻²·s⁻¹) | $L_s$ | $WUE$ (μmol·mmol⁻¹) |
|---|---|---|---|---|---|---|
| Mature trees | 1.530 ± 0.099 | 0.013 ± 0.003 | 223.328 ± 19.767 | 0.169 ± 0.037 | 0.472 ± 0.040 | 7.553 ± 0.804 |
| Seedlings | 1.161 ± 0.041 | 0.012 ± 0.002 | 301.949 ± 29.546 | 0.144 ± 0.028 | 0.294 ± 0.044 | 9.673 ± 1.083 |
| *P* value | <0.001** | 0.656 | 0.019* | 0.394 | 0.007** | 0.046* |

Notes:
The values in the table are the mean and standard deviation.
* Indicates significant difference.
** Indicates extremely significant difference.

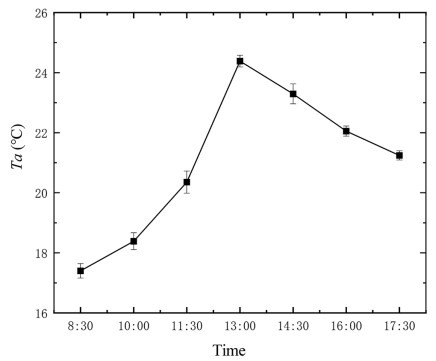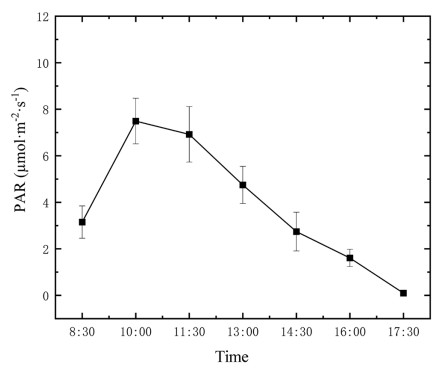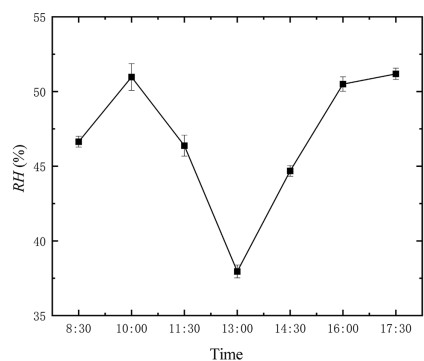

**Figure 2 Diurnal variation of environmental factors.**

potential for gas exchange, thereby enabling them to have a higher photosynthetic rate. The trend of $Tr$ was similar to that of $G_s$, indicating that transpiration is stronger under strong light conditions, thereby promoting the transport of internal water and inorganic salts. The $WUE$ of mature trees and seedlings initially increased and then decreased with increasing light intensity. The $WUE$ of mature trees reached its peak at 150 μmol·m⁻²·s⁻¹ and rapidly decreased thereafter. The $WUE$ of seedlings reached its peak at 200 μmol·m⁻²·s⁻¹ and slowly declined afterward. Mature trees exhibit higher $WUE$ under low light intensity, possibly related to their adaptation to harsh karst habitats. However, under high light intensity, $WUE$ shows a decreasing trend, indicating a weaker adaptation to high light intensity environments.

### Photosynthesis-light response parameters

Mature trees exhibited higher overall $Pn$ compared to seedlings, with significant differences observed in the maximum net photosynthetic rate ($P_{max}$) between mature trees (4.83 μmol·m⁻²·s⁻¹) and seedlings (3.85 μmol·m⁻²·s⁻¹) ($P < 0.05$), indicating that mature trees have stronger photosynthetic capabilities than seedlings and a greater ability to accumulate energy. Seedlings had significantly higher light compensation point (LCP) and dark respiration rate ($R_d$) ($P < 0.05$), measuring 4.70 μmol·m⁻²·s⁻¹ and 0.50 μmol·m⁻²·s⁻¹, respectively. The light saturation point (LSP) of mature trees was significantly higher ($P < 0.05$) at 1,185.31 μmol·m⁻²·s⁻¹. However, there were no significant differences ($P > 0.05$) in apparent quantum yield (AQY) between mature trees and seedlings (Table 2).

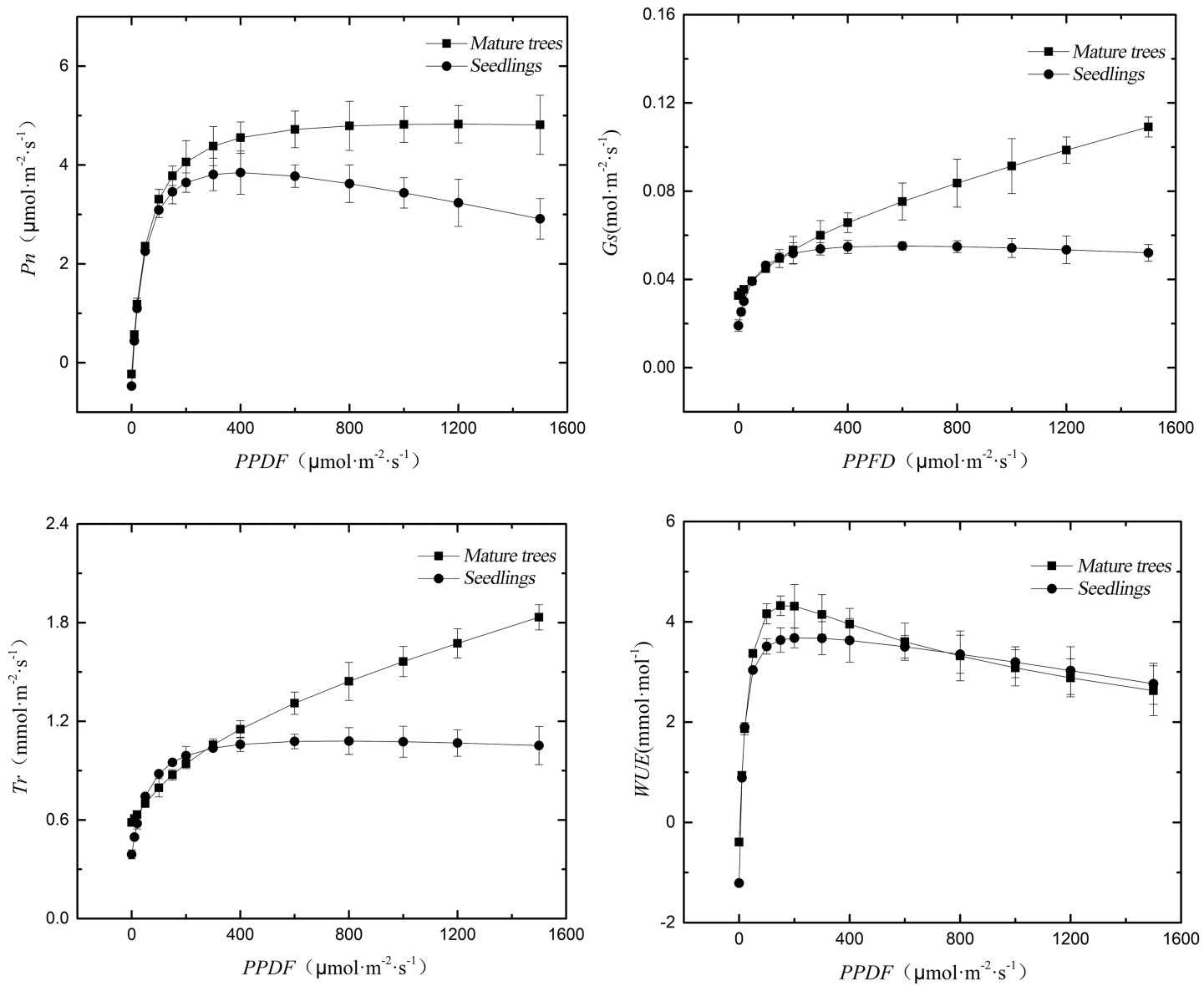

**Figure 3 Response of photosynthetic parameters in leaves of mature trees and seedlings of *H. hainanensis* to light intensity.**

**Table 2 Photosynthesis-light response parameters of leaves in mature trees and seedlings of *H. hainanensis*.**

| Type | $P_{max}$ (µmol·m⁻²·s⁻¹) | LCP (µmol·m⁻²·s⁻¹) | LSP (µmol·m⁻²·s⁻¹) | AQY (µmol·µmol⁻¹) | $R_d$ (µmol·m⁻²·s⁻¹) |
|---|---|---|---|---|---|
| Mature trees | 4.83 ± 0.16 | 2.69 ± 0.28 | 1,185.31 ± 23.65 | 0.025 ± 0.002 | 0.25 ± 0.07 |
| Seedlings | 3.85 ± 0.29 | 4.70 ± 0.44 | 403.45 ± 34.47 | 0.025 ± 0.007 | 0.50 ± 0.09 |
| *P* value | 0.015* | 0.005** | <0.001** | 0.448 | 0.038* |

Notes:
The values in the table are the mean and standard deviation.
* Indicates significant difference.
** Indicates extremely significant difference.

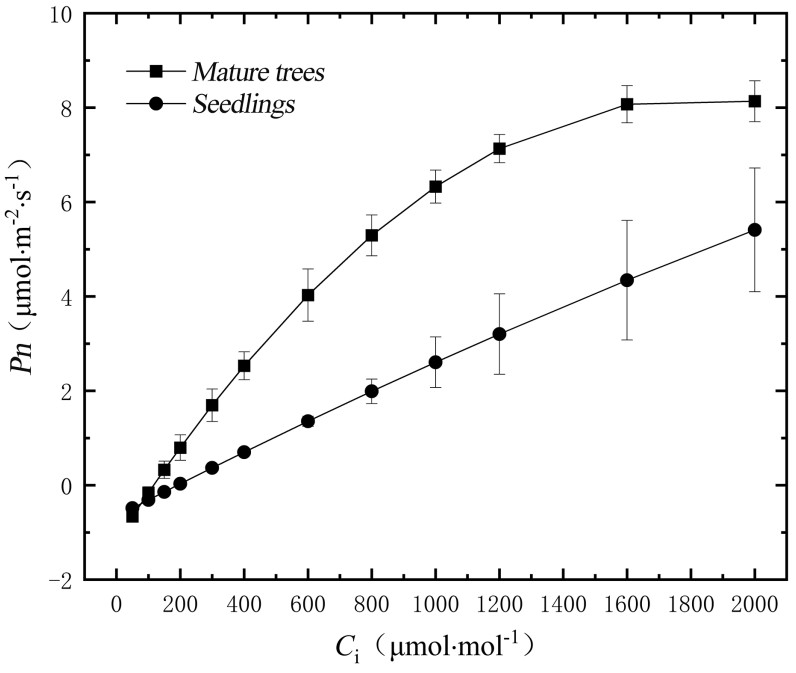

**Figure 4 Photosynthesis-CO$_2$ response curves of leaves in mature trees and seedlings of *H. hainanensis*.**

**Table 3 Photosynthesis-CO$_2$ response parameters of leaves in mature trees and seedlings of *H. hainanensis*.**

| Type | α (μmol·m$^{-2}$·s$^{-1}$) | $A_{max}$ (μmol·m$^{-2}$·s$^{-1}$) | CCP (μmol·m$^{-2}$·s$^{-1}$) | CSP (μmol·m$^{-2}$·s$^{-1}$) | $R_p$ (μmol·m$^{-2}$·s$^{-1}$) |
|---|---|---|---|---|---|
| Mature trees | 0.010 ± 0.001 | 8.214 ± 0.382 | 117.756 ± 21.054 | 1,829.814 ± 34.317 | 1.179 ± 0.133 |
| Seedlings | 0.004 ± 0.001 | 5.822 ± 0.259 | 253.453 ± 23.528 | 2,438.832 ± 76.247 | 0.852 ± 0.069 |
| *P* value | 0.002** | <0.001** | 0.002** | <0.001** | 0.019* |

**Notes:**
The values in the table are the mean and standard deviation.
* Indicates significant difference.
** Indicates extremely significant difference.

## Photosynthesis-CO$_2$ response parameters

The determination coefficients ($R^2$) for the fitting of the photosynthesis-CO$_2$ response curves in mature trees and seedlings of *H. hainanensis* were both above 0.95, indicating a good fitting effect (Fig. 4). With the increase of $C_i$, the *Pn* of mature trees showed an initial increase followed by a flattening trend, while the *Pn* of seedlings exhibited a continuous increase. After $C_i$ reached 100 μmol·mol$^{-1}$, the *Pn* of mature trees surpassed that of seedlings, and the final performance of $P_{max}$ showed mature trees > seedlings. According to Table 3, the initial carboxylation efficiency (α), potential maximum net photosynthetic rate ($A_{max}$), and light respiration rate ($R_p$) of mature trees were all higher than those of seedlings, with $R_p$ showing significant differences ($P < 0.05$) and α, $A_{max}$ showing extremely significant differences ($P < 0.01$). The CO$_2$ compensation point (CCP) and CO$_2$ saturation point (CSP) of seedlings were both higher than those of mature trees, and both showed extremely significant differences ($P < 0.01$).
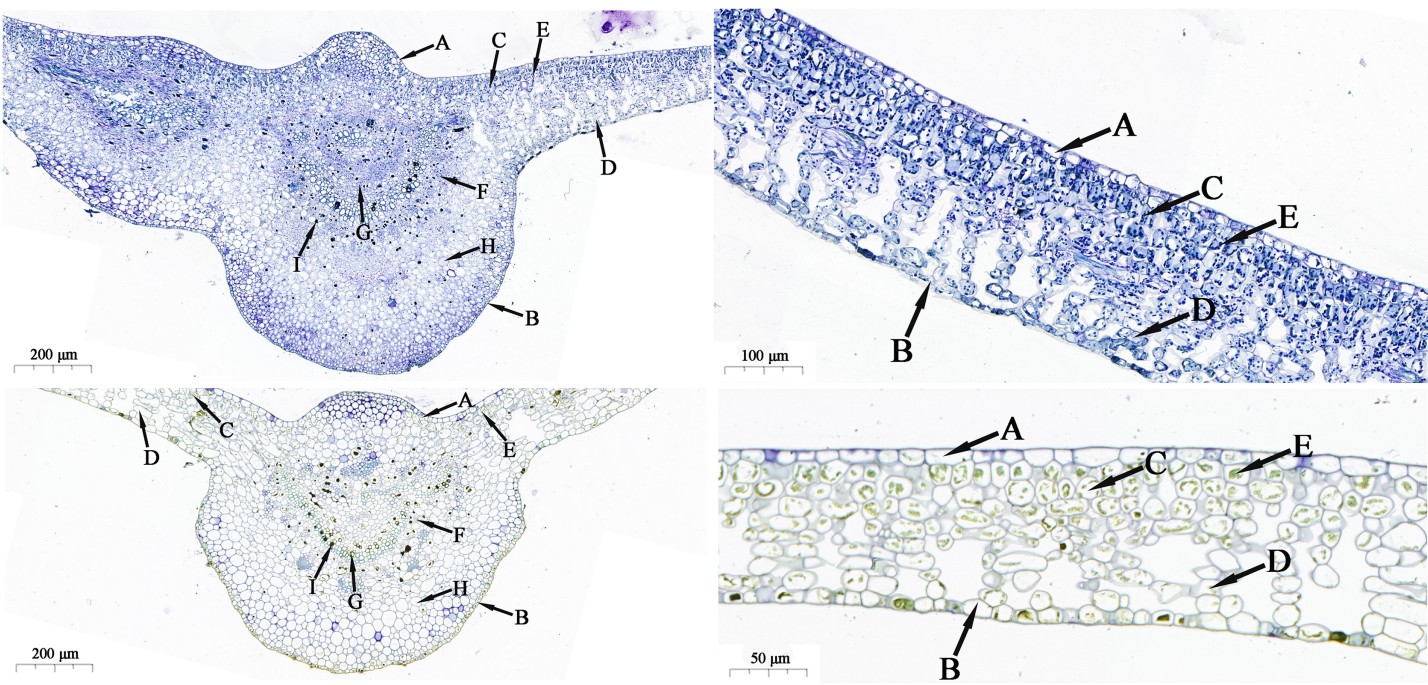

**Figure 5** Leaf structure of *H. hainanensis* in mature trees and seedlings (top: mature tree leaf; bottom: seedling leaf). UE, Upper epidermal; LE, Lower epidermal; PT, Palisade tissue; ST, Spongy tissue; Ch, Chloroplasts; VB, Vascular bundles; X, Xylem; AT, Aqueous tissue; AC, Abnormal cells.

**Table 4** Anatomical parameters of leaf structure in mature trees and seedlings of *H. hainanensis*.

| Index | Mature trees | Seedlings | *P* value |
|---|---|---|---|
| Leaf thickness (LT, μm) | 319.567 ± 0.939 | 253.300 ± 2.642 | <0.001** |
| Upper epidermis thickness (UET, μm) | 21.633 ± 0.946 | 24.533 ± 0.736 | 0.027* |
| Lower epidermis thickness (LET, μm) | 14.700 ± 0.245 | 24.367 ± 0.822 | <0.001** |
| Palisade parenchyma thickness (PPT, μm) | 139.300 ± 0.909 | 100.767 ± 1.852 | <0.001** |
| Spongy parenchyma thickness (SPT, μm) | 138.567 ± 1.731 | 110.567 ± 1.725 | <0.001** |
| PPT/SPT | 1.005 ± 0.009 | 0.911 ± 0.031 | 0.014* |
| Relative palisade parenchyma thickness (RPPT) | 0.436 ± 0.002 | 0.398 ± 0.003 | <0.001** |
| Relative spongy parenchyma thickness (RSPT) | 0.434 ± 0.004 | 0.437 ± 0.011 | 0.741 |

**Notes:**
The values in the table are the mean and standard deviation.
* Indicates significant difference.
** Indicates extremely significant difference.

## Foliar morphological characteristics

The cross-sectional structural morphological characteristics of leaf blades in mature trees and seedlings of *H. hainanensis* are shown in Fig. 5. The leaf blades are composed of upper epidermis, lower epidermis, palisade tissue, spongy tissue, and other differentiated cells. According to Table 4, the leaf thickness (LT), palisade parenchyma thickness (PPT), spongy parenchyma thickness (SPT), palisade-to-spongy ratio (PPT/SPT), and relative palisade parenchyma thickness (RPPT) of mature trees were significantly higher than those of seedlings ($P < 0.05$), with LT, PPT, SPT, and RPPT showing extremely significant
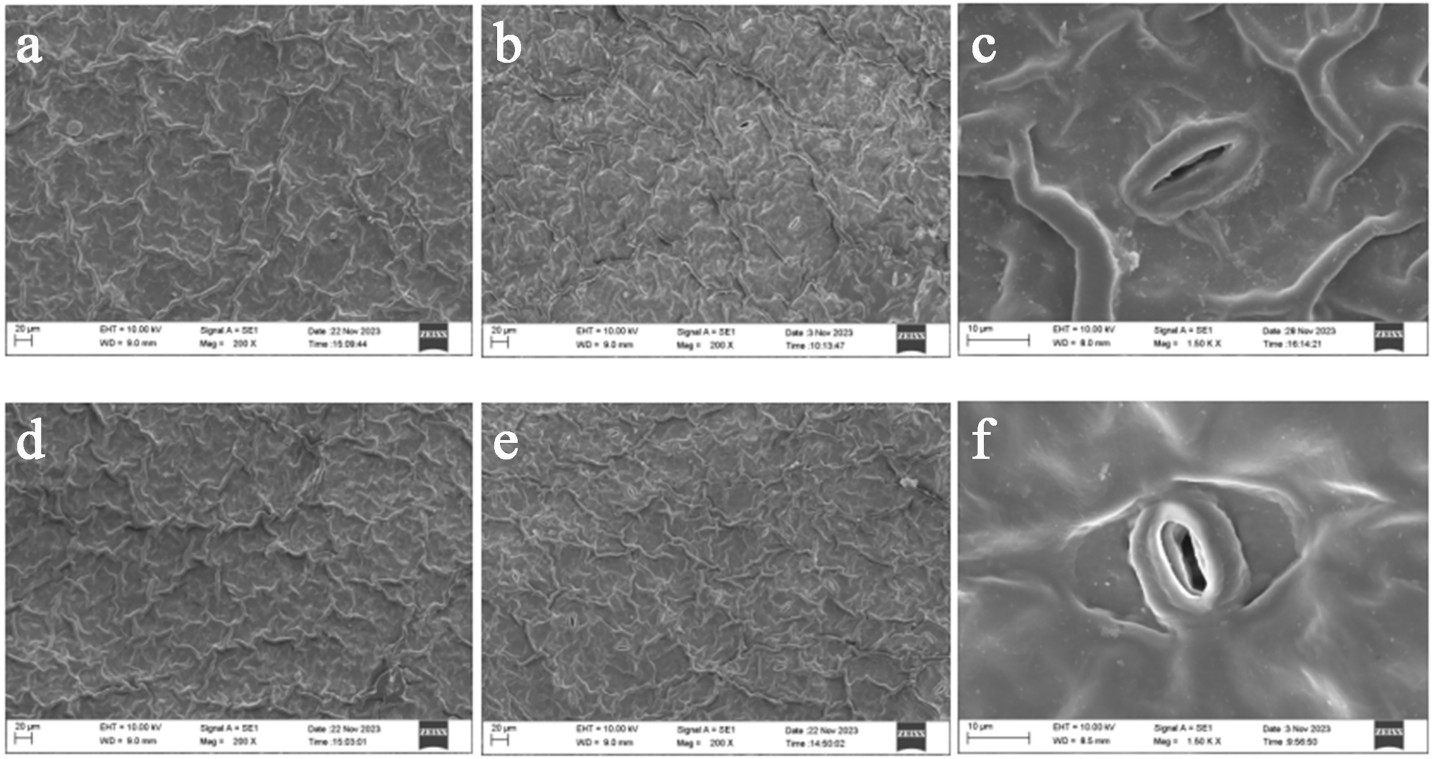

**Figure 6 The stomatal characteristics of mature trees (top) and seedling leaves (bottom) of *H. hainanensis*.**

differences ($P < 0.01$). The upper epidermis thickness (UET), lower epidermis thickness (LET), and relative spongy parenchyma thickness (RSPT) of seedlings were higher than those of mature trees, with significant differences observed in UET ($P < 0.05$) and extremely significant differences in LET ($P < 0.01$). However, there were no significant differences in RSPT ($P > 0.05$), indicating that the leaves of mature trees are better developed and have a stronger ability to photosynthesize.

## Leaf stomatal characteristics

Based on the leaf stomatal characteristics of mature trees (Figs. 6A–6C) and seedling (Figs. 6D–6F) leaves of *H. hainanensis*, it can be observed that, from left to right, they respectively represent the upper and lower epidermis and stomatal structures of the leaves. The leaf stomata are mainly distributed on the lower epidermis, and both the stomata and their guard cells appear elongated, primarily to avoid direct sunlight and reduce water loss. As indicated in Table 5, the average values of stomatal density, stomatal width, and individual stomatal area for seedlings are all greater than those of mature trees, the average values of stomatal width and individual stomatal area showing extremely significant differences ($P < 0.01$), the average values of stomatal density showing significant differences ($P < 0.05$), while the average values of stomatal length exhibits no significant difference ($P > 0.05$).

**Table 5 The stomatal parameters of mature trees and seedling leaves of *H. hainanensis*.**

| Type | Stomatal density (SD) (counts/mm$^2$) | Stomatal length (SL) (μm) | Stomatal width (SW) (μm) | Stomatal area (SA) (μm$^2$) |
|---|---|---|---|---|
| Mature trees | 95.26 ± 5.35 | 14.57 ± 1.30 | 3.24 ± 0.91 | 37.34 ± 12.15 |
| Seedlings | 106.23 ± 3.66 | 14.04 ± 0.35 | 6.93 ± 0.37 | 76.45 ± 5.95 |
| *P* value | 0.031[*] | 0.533 | 0.003[**] | 0.007[**] |

Notes:
The values in the table are the mean and standard deviation.
[*] Indicates significant difference.
[**] Indicates extremely significant difference.

**Table 6 Leaf photosynthetic pigment content and ratio in mature trees and seedlings of *H. hainanensis*.**

| Type | Chl a (mg·g$^{-1}$) | Chl b (mg·g$^{-1}$) | Chl (a+b) (mg·g$^{-1}$) | Car (mg·g$^{-1}$) | Chl a/b | Car/Chl (a+b) |
|---|---|---|---|---|---|---|
| Mature trees | 15.284 ± 1.243 | 11.314 ± 0.837 | 26.598 ± 2.314 | 3.250 ± 0.266 | 1.351 ± 0.213 | 0.122 ± 0.008 |
| Seedlings | 7.444 ± 0.773 | 4.206 ± 0.355 | 11.650 ± 1.147 | 2.323 ± 0.159 | 1.770 ± 0.140 | 0.199 ± 0.021 |
| *P* value | <0.001[**] | <0.001[**] | <0.001[**] | 0.005[**] | 0.045[*] | 0.004[**] |

Notes:
The values in the table are the mean and standard deviation.
[*] Indicates significant difference.
[**] Indicates extremely significant difference.

## Leaf photosynthetic pigment content and ratio

According to Table 6, it can be observed that the Chl a, Chl b, Chl (a+b), and Car content of mature trees of *H. hainanensis* are higher than those of seedlings. However, the Chl a/b ratio and Car/Chl (a+b) ratio of seedlings are higher than those of mature trees. Among them, Chl a, Chl b, Chl (a+b), Car, and Car/Chl (a+b) show extremely significant differences ($P < 0.01$), while Chl a/b ratio shows a significant difference ($P < 0.05$).

## Correlation analysis of photosynthetic physiological parameters, leaf structural characteristics, and photosynthetic pigment content

Figure 7 shows the heatmap of the correlation analysis among photosynthetic physiological parameters, leaf structural characteristics, and photosynthetic pigment content in mature trees and seedlings of *H. hainanensis*. From Fig. 7, it can be observed that there is a strong correlation among photosynthetic physiological parameters, leaf structural characteristics, and photosynthetic pigment content in both mature trees and seedlings. Specifically, the photosynthetic pigments Chl a, Chl b, Chl (a+b), and Car show significant positive correlations with $P_{max}$, LSP, α, $A_{max}$, $R_p$, Pn, Ls, LT, PPT, SPT, PPT/SPT, and TT. They also show significant negative correlations with CCP, CSP, and LET. Stomatal structural parameters SD, SW, and SA exhibit significant positive correlations with LCP, $R_d$, CCP, CSP, $C_i$, *WUE*, UET, and LET. Leaf structural parameters UET and LET show significant positive correlations with LCP, $R_d$, CCP, CSP, $C_i$, and *WUE*, while they exhibit significant negative correlations with PPT, SPT, and TT. Moreover, PPT, SPT, PPT/SPT, and TT show significant positive correlations with $P_{max}$, LSP, α, $A_{max}$, $R_p$, Pn, Ls, and LT. Photosynthetic physiological parameters $R_p$, Pn, and Ls show significant positive correlations with $P_{max}$, LSP, α, and $A_{max}$. α and $A_{max}$ show significant positive correlations

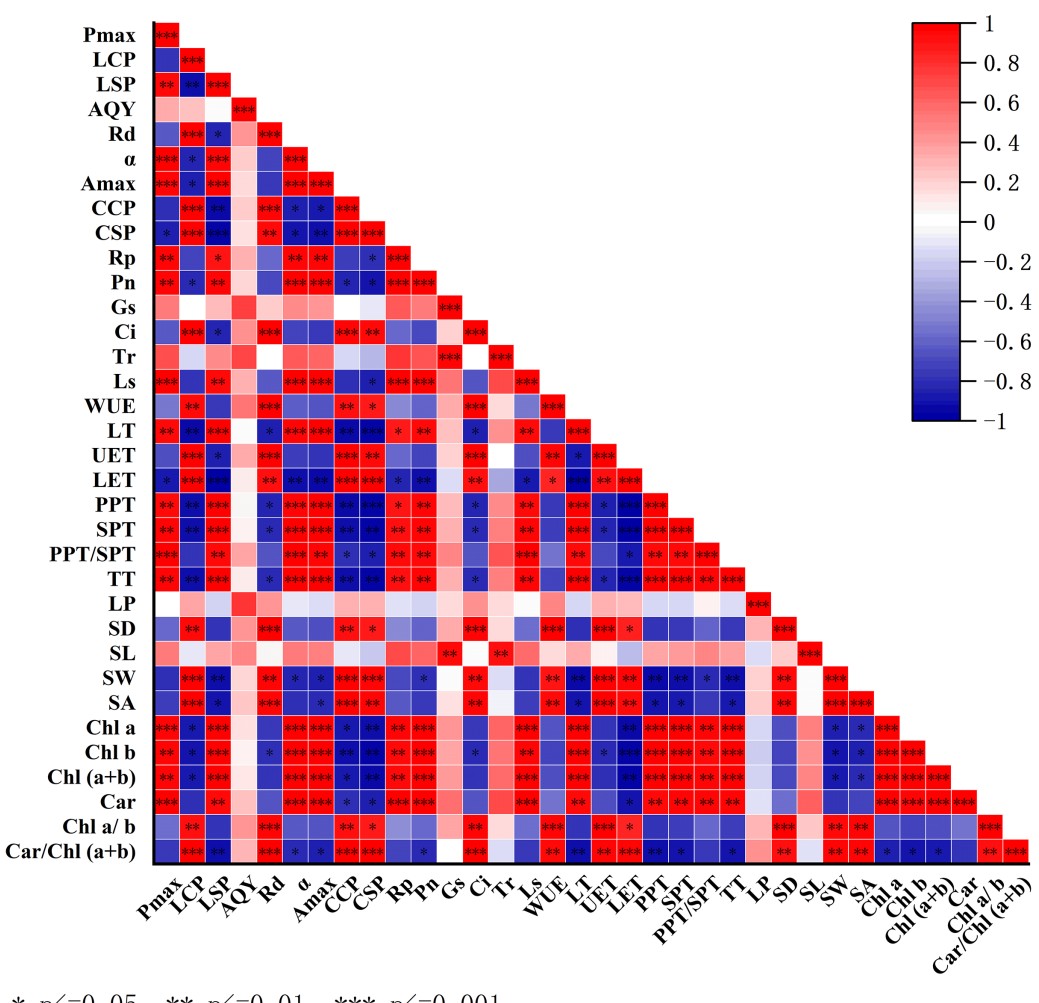

* p<=0. 05    ** p<=0. 01    *** p<=0. 001

**Figure 7 Heatmap of correlation analysis among photosynthetic physiological parameters, leaf structural characteristics, and photosynthetic pigments in mature trees and seedlings of *H. hainanensis*.**

with $P_{max}$ and LSP. CCP and CSP exhibit significant positive correlations with LCP and $R_d$, while showing significant negative correlations with LSP, α, and $A_{max}$.

## DISCUSSION

### Photosynthetic characteristics

By calculating the light and parameters of plants, further exploration of their suitable growth environments and their abilities to adapt to the environment can be conducted (*Yokoya et al., 2007*). A higher maximum net photosynthetic rate ($P_{max}$) indicates a stronger carbon fixation ability and a more abundant accumulation of organic matter, making it easier to meet the organic material requirements for plant growth (*Mahmud et al., 2018*; *Vona et al., 2018*). The $P_{max}$ of mature trees of *H. hainanensis* (4.83 µmol·m$^{-2}$·s$^{-1}$) is significantly higher than that of seedlings (3.85 µmol·m$^{-2}$·s$^{-1}$) (Table 2), indicating that the photosynthetic capacity of mature trees is significantly higher than that

of seedlings, and mature trees have a higher ability to accumulate organic matter compared to seedlings. One possible reason is that the organic matter required to sustain the normal growth of mature trees is greater than that of seedlings, so mature trees need a higher light utilization capacity to maintain normal physiological activities. To some extent, the palisade parenchyma thickness and chlorophyll content of mature tree leaves are significantly greater than those of seedlings.

The LCP and LSP reflect a plant's adaptation and utilization abilities towards light, serving as important indicators for evaluating plant growth characteristics. A larger difference between LCP and LSP indicates a wider range of light adaptation for the plant (*Zhang et al., 2005*). It is generally believed that shade-tolerant plants have an LCP below 20 $\mu mol \cdot m^{-2} \cdot s^{-1}$ and an LSP of 500–1,000 $\mu mol \cdot m^{-2} \cdot s^{-1}$ or lower (*Jiang, 2004*). The results of this study indicate that the LCP of mature trees is 2.69 $\mu mol \cdot m^{-2} \cdot s^{-1}$, and the LSP is 1,185.31 $\mu mol \cdot m^{-2} \cdot s^{-1}$. The LCP of seedlings is 4.70 $\mu mol \cdot m^{-2} \cdot s^{-1}$, and the LSP is 403.45 $\mu mol \cdot m^{-2} \cdot s^{-1}$ (Table 2). These values demonstrate typical characteristics of shade-tolerant plants, indicating that *H. hainanensis* seedlings require a certain degree of shading for growth. The difference between LCP and LSP in mature trees is greater than that in seedlings, with mature plants having a higher LCP and a lower LSP compared to seedlings, suggesting that mature plants have a stronger adaptability to light environments than seedlings. Similar observations have been made in other plants such as *Argentina anserina*, *Populus*, and *Bletilla striata* (*Li et al., 2022*; *Chen et al., 2022*; *Luo et al., 2019*). *Sheng et al. (2020)* found that larger stomatal apertures are one of the characteristics enabling plants to adapt to low light conditions. According to the correlation analysis in this study, stomatal conductance shows a strong correlation with photosynthetic-light response parameters, with seedlings exhibiting significantly larger stomatal conductance than mature trees, consistent with the previous findings of seedlings adapted to low light environments. Therefore, in the cultivation of *H. hainanensis*, it is recommended to provide certain shading treatment during the seedling stage and gradually reduce the shading as the plants grow.

$CO_2$ is one of the primary raw materials for photosynthesis in plants. Sufficient $CO_2$ can promote the net photosynthetic rate of plants, while insufficient $CO_2$ can inhibit the accumulation of organic compounds. As $CO_2$ concentration gradually increases, it can enhance carboxylase activity while inhibiting ribulose-1,5-bisphosphate carboxylase, thereby enhancing photosynthesis (*Li et al., 2016*; *Sun et al., 2010*). The $A_{max}$ of mature trees is significantly higher than that of seedlings (Table 3), indicating that mature trees have a higher capacity to utilize $CO_2$ compared to seedlings. The $\alpha$ of seedlings is significantly higher than that of mature trees, suggesting that seedlings have a stronger adaptation ability in a low $CO_2$ environment. Meanwhile, the range of CSP is relatively large, indicating that *H. hainanensis* has a broad range of adaptability to $CO_2$ concentrations. Due to the lower $P_{max}$ compared to $A_{max}$, insufficient $CO_2$ supply in the environment is one of the main limiting factors for its photosynthetic rate. Therefore, planting mature trees and seedlings of *H. hainanensis* in an environment with higher $CO_2$ concentration is beneficial for their growth and development. Stomata, unique structures on the leaf surface, control gas exchange and water loss in leaves. According to the

correlation analysis in this study, there is a significant correlation between stomatal density, stomatal conductance, CCP, CSP, and $C_i$. Both stomatal density and stomatal conductance are higher in seedlings compared to mature trees, indicating that seedlings have a stronger ability for gas exchange and are more efficient in absorbing more $CO_2$ in a low $CO_2$ environment, demonstrating their stronger adaptability to such conditions, consistent with the previous results. However, due to factors such as higher chlorophyll content and more mature development of photosynthetic organs in mature trees, they have a stronger capacity to utilize $CO_2$, resulting in generally higher net photosynthetic rates (*Kinose et al., 2020*).

During the diurnal variation of photosynthesis (Fig. 1), as the Beijing time approaches noon, the light intensity becomes excessively high, the environmental temperature gradually increases while humidity decreases. This leads to enhanced transpiration in plant leaves. Faced with environmental water scarcity and rapid internal water loss, the leaves opt to reduce stomatal conductance or even close stomata to decrease transpiration, reduce water loss, and simultaneously decrease $CO_2$ uptake. This results in both mature trees and seedlings displaying low values of *Pn*. When the "midday depression" phenomenon occurs, $C_i$ shows an upward trend, by adjusting *Gs* to fix the $CO_2$ level in the environment to the maximum extent and further reducing leaf water loss, the impact of high temperature can be mitigated. Similar observations were made by *Xu et al. (2024)* in three species of Geodorum plants. We found that PAR may influence environmental factors such as temperature and humidity. Around 10:00 a.m., light intensity gradually increases, while temperature and humidity enter suitable ranges. This adjustment modulates *Gs*, promoting stomatal opening and gas exchange, raising leaf temperature, increasing the vapor pressure difference inside and outside the leaf, thus accelerating transpiration rate, ultimately enhancing photosynthetic rate. Subsequently, as PAR decreases, environmental temperature and humidity exceed optimal ranges, causing leaf *Gs* to decline gradually, thus reducing the photosynthetic rate. The average daily net photosynthetic rate of mature trees is significantly higher than that of seedlings, indicating that seedlings may face difficulties in accumulating substances and experience developmental constraints. This puts them at a disadvantage in intense population competition, resulting in slow population renewal and gradually entering a "death spiral." *Jiang (2018)* also found that weak adaptability and low survival rate of *H. hainanensis* seedlings are among the main reasons for their endangered status. Similar phenomena have been observed in endangered plants such as *Vatica guangxiensis* (*Xiao et al., 2023*), *Abies ziyuanensis* (*Li, 2024*), and *Glyptostrobus pensilis* (*Zheng, 2021*).

## The relationship between leaf anatomy and photosynthetic characteristics

This study found both upper and lower epidermises consist of a single layer of cells. In mature trees, most of the cells in the upper and lower epidermis are nearly rectangular, while in seedlings, they are irregularly elliptical. The palisade tissue in mature trees consists of 5–7 layers of cells, mostly rectangular in shape, arranged in an orderly and compact manner. The palisade tissue in seedlings comprises 4–5 layers of cells, which are irregularly

elliptical, loosely arranged with some intercellular spaces, and the cells are shorter. In comparison to seedlings, the spongy tissue in mature trees exhibits a denser arrangement, more layers, and smaller intercellular spaces. Mature trees have well-developed vascular bundles, with 5–7 rows of conduits in the xylem. Abundant heterogeneous cells are mainly distributed near the vascular bundles and the xylem. The water storage tissue cells are small in size, numerous, and densely arranged. In contrast, the vascular bundles in seedlings are less developed, with 4–6 rows of conduits in the xylem. There are fewer heterogeneous cells compared to mature trees, primarily distributed near the vascular bundles and the xylem. The water storage tissue cells are larger in size and fewer in number. The parenchyma tissue cells are larger and plump, with some cells differentiating into water storage tissue, providing moisture for normal physiological activities of the seedlings (Fig. 5). The leaf is one of the vital organs for photosynthesis in plants, and the structural parameters of the UET, LET, PPT and SPT in leaves can all be influenced by external environmental factors, thereby altering the plant's photosynthetic characteristics. The cellular structural features of leaves adapted to low light conditions include smaller upper and lower epidermis thickness, larger leaf thickness, and well-developed palisade and spongy tissues (Li et al., 2020a; Chen et al., 2020). In this study, the leaf thickness of mature trees was significantly greater than that of seedlings, while the thickness of both the upper and lower epidermis was significantly lower than that of seedlings. Additionally, both the palisade and spongy parenchyma thicknesses were significantly higher in mature trees compared to seedlings (Table 4), indicating that mature trees are better adapted to growth under low light environments. The palisade-to-spongy tissue ratio is an indicator reflecting the development of palisade parenchyma in plant leaves, where a higher value indicates more developed palisade parenchyma (Liang et al., 2014). Moreover, plant photosynthesis is significantly influenced by the palisade-to-spongy tissue ratio (Dong et al., 2022). A tightly arranged palisade parenchyma with cells closer to a rectangular shape can effectively increase the distribution density of chloroplasts within palisade cells, while a greater thickness of spongy parenchyma with a looser arrangement is favorable for gas exchange and photosynthesis in leaves (Xue, 2020). In mature trees of H. hainanensis, the palisade parenchyma cells are mostly rectangular in shape compared to seedlings, and the palisade-to-spongy tissue ratio is significantly higher, indicating that the palisade parenchyma in mature trees is more developed, with significantly higher RPPT than in seedlings. Consequently, the arrangement is more compact, and chloroplast distribution within palisade parenchyma cells is more concentrated, indicating that the photosynthetic activity of mature H. hainanensis trees is stronger than that of seedlings.

Research has shown that plant photosynthesis is highly sensitive to arid environments, primarily affecting plant photosynthesis by inhibiting the photosynthetic system and reducing the content of photosynthetic enzymes (Liu et al., 2023). This study found that both types of leaves have a higher number of abnormal cells surrounding leaf veins, while the distribution of upper and lower epidermis is relatively sparse (Fig. 5). Abnormal cells have a higher osmotic potential and stronger water absorption capacity, allowing them to store water. When facing a drought environment, they can at least temporarily provide

water to other cells, enhancing resistance (*Yang et al., 2007*). The vascular bundle is one of the important factors influencing the efficiency of water transport in plants. The more developed the vascular bundle, the stronger the efficiency of water transport, and the degree of vascular bundle development can reflect the plant's drought resistance (*Ma et al., 2020*). The xylem is mainly responsible for the transport of water and inorganic salts. The more developed the xylem, the better the plant's water and inorganic salt transport (*Li et al., 2023*). The content of abnormal cells in mature trees is higher than in seedlings, and the vascular bundles and xylem are well-developed, indicating abundant water storage cells. Therefore, it can be inferred that mature trees have stronger drought resistance than seedlings, and similar phenomena have been observed in plants such as *Populus euphratica* and *Populus pruinosa* (*Zhao, 2016*). Even in arid environments, leaves are still able to temporarily provide water and carry out normal photosynthesis to provide energy for themselves.

## The relationship between photosynthetic pigments and photosynthetic characteristics

Chlorophyll is one of the essential components of photosynthetic pigments in plants, and its content and ratio are key indicators of a plant's adaptive capacity to the environment (*Rodriguez-Calcerrada et al., 2008*). The Chl a/b ratio of mature *H. hainanensis* trees and seedlings is consistently below 3 (Table 6), indicating shade-tolerant characteristics. However, mature trees exhibit higher chlorophyll content and lower Chl a/b ratio, suggesting their enhanced utilization of light energy (*Huang et al., 2016*; *Hoflacher & Bauer, 1982*). Correlation analysis results demonstrate a significant positive relationship between chlorophyll and *Pn*, as well as leaf structure. Based on the actual growth conditions, it is speculated that seedlings generally occupy lower positions within the community, where abundant light is intercepted by mature trees and other tall plants. In response to low light conditions, seedlings may adapt by reducing chlorophyll content to avoid photoinhibition caused by excessive chlorophyll levels. However, in actual community environments, excessive shading of seedlings often results in insufficient light absorption, exacerbating cellular and tissue underdevelopment and chlorophyll deficiency, leading to decreased *Pn* and forming a vicious cycle. These findings align with the diurnal variations in photosynthesis and leaf structure mentioned earlier. *Pan et al. (2024b)* also discovered similar survival challenges among seedlings of *Vatica guangxiensis*.

## CONCLUSIONS

In summary, there are differences in the photosynthetic characteristics between mature *H. hainanensis* trees and seedlings, and these differences are primarily attributed to variations in photosynthetic pigments and leaf structure. While mature trees exhibit a broader range of light adaptability, they are more suited to grow under low light intensity. Seedlings, on the other hand, display typical shade-tolerant characteristics and are only suitable for growth in environments with high shading and sufficient moisture. The underdeveloped photosynthetic organs and poor adaptability in seedlings, insufficient

natural light and low survival rates during wild growth are likely one of the reasons for *H. hainanensis* endangered status. Therefore, during in-site conservation, it is possible to promote the growth and development of seedlings by appropriately employing artificial "windowing" or shading methods based on the actual growth environment of the seedlings. It is recommended to provide appropriate shade for seedlings during artificial cultivation of *H. hainanensis*, while ensuring adequate water and $CO_2$ levels. As the plants grow, the degree of shading can be gradually reduced. Once the plants reach maturity, they can be transplanted to environments with lower shading. This study aims to provide a theoretical basis for the *in situ* conservation and artificial cultivation of *H. hainanensis* through fundamental research on the photosynthetic physiology, leaf structure, and photosynthetic pigments of mature trees and seedlings.

### Funding

This study was supported by the Subproject of National Key R&D Program (2022YFF1300703); the Guangxi Forestry Science and Technology Promotion Demonstration Project ([2022]GT23); and the Hechi City Science and Technology Plan Project (Hechi Science AC231113). The funders had no role in study design, data collection and analysis, decision to publish, or preparation of the manuscript.

### Grant Disclosures

The following grant information was disclosed by the authors:
Subproject of National Key R&D Program: 2022YFF1300703.
Guangxi Forestry Science and Technology Promotion Demonstration Project: [2022] GT23.
Hechi City Science and Technology Plan Project: AC231113.

### Competing Interests

The authors declare that they have no competing interests.

### Author Contributions

- Jianwang Xu conceived and designed the experiments, performed the experiments, analyzed the data, prepared figures and/or tables, and approved the final draft.
- Jianmin Tang conceived and designed the experiments, authored or reviewed drafts of the article, and approved the final draft.
- Haolong Jiang analyzed the data, prepared figures and/or tables, and approved the final draft.
- Rong Zou conceived and designed the experiments, analyzed the data, prepared figures and/or tables, authored or reviewed drafts of the article, and approved the final draft.
- Xiao Wei conceived and designed the experiments, authored or reviewed drafts of the article, and approved the final draft.

## Data Availability

The raw data is available in the Supplemental Files.

## Supplemental Information

Supplemental information for this article can be found online at http://dx.doi.org/10.7717/peerj.18640#supplemental-information.

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
