# Peer review of "A comparative study on leaf anatomy and photosynthetic characteristics of different growth stages of *Horsfieldia hainanensis"

_PeerJ, doi:10.7717/peerj.18640_

## Round 0.1 · original submission · Major Revisions

Two expert reviewers have evaluated your manuscript and their comments can be seen below. As you will see, both reviewers have positive remarks about your study, and they also have a number of suggestions to improve the manuscript. Please ensure that you address all of these comments in a rebuttal that clearly indicates where and how you have modified the manuscript or provide justification where you have not.

·

Basic reporting

This study addresses an interesting research area concerning the capacity of plant species to respond to their environments based on their photosynthetic systems and related mechanisms. It provides valuable insights into the responses of Horsfeldia hainanensis at different growth stages. The study is worth considering for publication, though several issues need to be addressed prior to acceptance.
First, I recommend the authors seek professional English editing to improve the clarity of the content and the logical flow of ideas.
Second, the introduction should be expanded to include recent findings on the effects of growth form on photosynthesis, placing the research within a broader scientific context. The study's objectives should also be revised to better reflect the findings.
Lastly, I encourage the authors to take my general comments into consideration.

Experimental design

The experimental design is generally sound, though some aspects require further attention, as detailed in the General Comments section

Validity of the findings

The results are clearly presented and convey the study's key messages effectively. Some minor revisions are required (see comments below).
However, the supplementary data provided are insufficient for understanding the analysis, as it is unclear how the values correspond to the growth stages or leaf types. Please revise this section with clearer descriptions of the variables.

Additional comments

Data analysis: Why did the authors not use multivariate analysis, such as PCA, to highlight the variability trends between trees and seedlings based on the studied variables?
Discussion: The first paragraph of the discussion merely summarizes the results. I suggest the authors begin by discussing the results directly.
Figures 1, 2, and 3: The quality of these figures needs improvement. Please provide higher-resolution versions.
Figure 5: The scale is not visible, and the abbreviations need to be clearer. I suggest using separate numbering for each part (e.g., A, B, etc.). Additionally, there is a notable difference in the tissue colors. Was this due to different chemical concentrations used for mature and seedling plants?
Tables 1-5: Please indicate which variables show significant differences by, for example, using asterisks for p-values. Clarify whether the values represent means and standard deviations or other metrics.
Table 4: Ensure that there is space between the variables and abbreviations.
Table 5: Please revise the unit of stomatal density.
Figure 7: Consider making the dots black for better clarity.

Minor Comments:
• Line 18: Please revise; the meaning is unclear.
• Line 56: Revise the phrase "the scientific research that makes the provided description possible."
• Line 59: Consider revision, as this line feels disconnected from the main message.
• Line 64: Include recent references on both trees and herbaceous plants.
• Lines 64-65: Revise and support with additional references.
• Line 72: Revise the connection with anatomical structure.
• Lines 77-79: Clearly state the objectives, rather than simply listing the variables measured.
• Lines 86-93: Provide references to support your statements, as this is important for future research in this area.
• Line 96: Were the soils used for the seedlings the same as those for the mature trees? Please specify.
• Line 97: Specify whether commercial or custom soil mixtures were used.
• Line 97: Are the seedlings derived from the mature trees used in the study or from another source? Please clarify.
• Line 111: Revise, as there is only one species under study.
• Line 136: Were the same plants used for the photosynthesis-light response curves? How many growth stages were studied? Were the same three plants used at each growth stage? Please clarify.
• Line 138: Did you use the portable Li-6400 photosynthesis system or another instrument? Please specify.
• Line 160: Clarify whether these plants were also used for measuring other parameters.
• Line 185: Explain why both sides were measured.
• Line 212: Specify the type of analysis and which parameters were calculated in Excel.
• Line 213: Given the comparison between two groups (trees and seedlings), why was ANOVA used?
• Lines 217-218: This sentence may be better suited to the "Measurement of Leaf Stomatal Structure" section.
• Line 241: Italicize "Ta."
• Line 282: Present the results directly. I suggest removing lines 285 to 301 and integrating them into the discussion.
• Line 312: Use appropriate figure subtitles (e.g., Figure 6a, etc.).
• Line 319: Include mean values to illustrate the significant differences observed in most stomatal traits.

Reviewer 2 ·

Basic reporting

Abstract
Line 21 You don’t need to mention your equipment in your abstract, that’s perfect for your Materials and Methods.

Introduction
General comment: I don’t feel like this section provides the background a reader needs. This should be expanded and contain more information. Maybe build it up from a problem, provide some insights (it can also be from other species), and then focus on why you will test everything on H. hainanensis. There are other good options, you can do what you want, but the introduction needs to be improved.

Experimental design

Materials and Methods
Line 180 I don’t think compactness is the right term. Also, you are only considering the palisade, so it should not be termed “tissue”, maybe something like “Relative palisade parenchyma thickness”
Line 180-181 This is not porosity. In terms with my previous comment, this would be “Relative spongy parenchyma thickness”. Porosity would be the volume of air in the leaf / total volume of the leaf
Line 198-209 Add a reference for these equations.

Validity of the findings

Results
Line 222-236 I wouldn’t talk about ‘M’ and ‘W’ trends. These trends are not clear, and you try to use them for all graphs which is definitely not the case, e.g.: Line 225 For Ls maybe, but not for WUE. Furthermore, did you compare these results with the abiotic conditions? I suspect a strong link between RH and Pn based on Figure 2. Maybe calculate VPD as well. Gs might correlate strongly with PAR, which would make sense.
Line 258-259 This is not significant based on your figures.
Line 286 It would be nice to have porosity here to provide numbers for this statement.
Line 293 This would also benefit from porosity values.
Line 315-317 You can’t say that based on these images, that is just how the stomata are fixed for your images.
Figure 7 This figure is not that clear. As the heat map is mirrored over the primary diagonal, you could remove the top right triangle. Move your legend below or above the matrix and make the matrix larger.

Discussion
Add references to your figures and tables so the reader knows where to look for the information.
Line 375-378 This also depends on total leaf area, right? If mature trees have the same leaf area, they need more efficient leaves; if they have a larger leaf area, they might not.
Line 407 Are all these digits significant?
Line 423-426 I don’t completely agree. So the midday depression is probably due to the environment, that’s true. Stomata could close due to an increase in VPD, which would limit Pn. I think Pn and Gs are correlated based on Figure 1.
Line 444-446 Did you take into account possible differences between shade and sun leaves?
Line 466 This sounds arbitrary. What are abnormal cells?
Line 489-491 As mentioned above, this could also explain differences in leaf thickness.

Conclusions
Line 507-512 This looks like a contradiction. First you say to provide more light to seedlings by creating a window, then you say to provide shade and increase the amount of light as the plants grow. Which is it?

Additional comments

I reviewed the manuscript ‘A comparative study on leaf anatomy and photosynthetic characteristics of different growth stages in Horsfieldia hainanensis’ by Xu et al. Overall, the introduction can be improved, and the analysis and discussion might be provided in more depth, but I do think the authors have performed some nice experiments. I have added my comments below. The authors should be able to address these comments based on the data they have acquired. Good luck!

[signed Jeroen D.M. Schreel]

---

## Round 0.2 · Minor Revisions

We have received comments back from one reviewer who remarks that most of the findings and suggestions were addreseed in the revised version. However, there are still issues with the use of the English language which need to be taken care of before the manuscript can be accepted for publication.

·

Basic reporting

I think that the authors satisfied most comments and suggestions that lead to a better version of the paper. Thus, the revisions made has improved the quality of the content.
Meanwhile, I think there is a room for improvement within the text. Please see the attached file and my comments within.

Minor comments:
Lines 50 – 60:
I still believe that there is a room for improvement. For instance, the first paragraph need some revision at both reformulation and English editing.
For instance, I could propose the following modification for more clarity and flow between ideas:
‘Horsfieldia hainanensis, a tall tree of the Myristicaceae family, is primarily found in Hainan and southern Guangxi, China. This species typically inhabits shady, moist forests in valleys and hills at elevations between 400 and 450 meters above sea level. It has been listed in both the "List of Extremely Small Population Wild Plants in Guangxi" and the "List of Key Protected Wild Plants in China." Due to its limited geographical distribution, population decline, and rarity, H. hainanensis is considered part of an extremely small population. Such populations are critical reservoirs of biological and genetic diversity. Without timely conservation efforts, their unique biological traits and genetic resources may be lost. As a result, protecting plants of extremely small populations is crucial for preserving biodiversity in China.’

Lines 68-71: This is an example, so you can began the sentence with : For instance, the related....

Line 105: Please change 'enjoys' woth 'benefit from
Are 'favorable conditions' refer to H. hainanesis ecological habitat? Pelase clarify.

Line 107: Is this the average or the accumulative rate? Please clarify if the accumulative precipitation.

Line 194: Please provide the full name of FAA

Line 208: here provide the total analyzed. Three leaves per plant was already explained.

Lines 209-211: Please merge this twi sentences for more clarity and flow.

Line 246: ‘with the passage of time’ I think this is not necessary, you can removed!

Line 361: I think this is not necessary, you can removed!

Experimental design

see my minor comments in section 1.

Validity of the findings

see my minor comments in section 1.

Additional comments

see my minor comments in section 1.

---

## Round 0.3 · Minor Revisions

The paragraph from lines 266 to 275 is written in the present instead of the past tense. Please see the attached PDF.

---

## Round 0.4 · accepted · Accept

Thank you for making the final changes. In my opinion the manuscript is now suitable for acceptance by PeerJ.